# Optimization-Based Algebraic Multigrid Coarsening Using Reinforcement Learning

**Ali Taghibakhshi**
Mechanical Science and Engineering
University of Illinois at Urbana-Champaign
Urbana, IL 61801, USA
alit2@illinois.edu

**Scott MacLachlan**
Mathematics and Statistics
Memorial University of Newfoundland
and Labrador
St. John's, NL, Canada
smaclachlan@mun.ca

**Luke Olson**
Computer Science
University of Illinois at Urbana-Champaign
Urbana, IL 61801, USA
lukeo@illinois.edu

**Matthew West**
Mechanical Science and Engineering
University of Illinois at Urbana-Champaign
Urbana, IL 61801, USA
mwest@illinois.edu

## Abstract

Large sparse linear systems of equations are ubiquitous in science and engineering, such as those arising from discretizations of partial differential equations. Algebraic multigrid (AMG) methods are one of the most common methods of solving such linear systems, with an extensive body of underlying mathematical theory. A system of linear equations defines a graph on the set of unknowns and each level of a multigrid solver requires the selection of an appropriate coarse graph along with restriction and interpolation operators that map to and from the coarse representation. The efficiency of the multigrid solver depends critically on this selection and many selection methods have been developed over the years. Recently, it has been demonstrated that it is possible to directly learn the AMG interpolation and restriction operators, given a coarse graph selection. In this paper, we consider the complementary problem of learning to coarsen graphs for a multigrid solver, a necessary step in developing fully learnable AMG methods. We propose a method using a reinforcement learning (RL) agent based on graph neural networks (GNNs), which can learn to perform graph coarsening on small planar training graphs and then be applied to unstructured large planar graphs, assuming bounded node degree. We demonstrate that this method can produce better coarse graphs than existing algorithms, even as the graph size increases and other properties of the graph are varied. We also propose an efficient inference procedure for performing graph coarsening that results in linear time complexity in graph size.

## 1 Introduction

Algebraic multigrid (AMG) [1, 2, 3] is a widely used method for approximating the solution to large, sparse linear systems, particularly those arising from elliptic boundary value problems. AMG has proven to be effective for a variety of problems, including partial differential equations (PDEs), image and video processing [4, 5], and sparse Markov chains [6]. The focus of this work is on developing reinforcement learning agents to construct optimal AMG methods for improved efficiency and robustness.

35th Conference on Neural Information Processing Systems (NeurIPS 2021).

AMG algorithms aim to solve a sparse linear system of the form

$$Ax = b \tag{1}$$

by successively constructing coarser representations of the problem. Constructing an AMG method is effectively a graph coarsening problem, to define a coarse representation of $A_c x_c = b_c$, and an interpolation problem, to define the transfer of solutions between the coarse and original (or fine) representation of the problem.

AMG has strong theoretical support for model problems [2, 7, 8], but it is important to recognize that the classical *algorithm* for AMG relies heavily on heuristics. Thus, there is a persistent need to both improve the general theoretical foundations of AMG [9, 10, 11, 12] and to improve its efficiency, including by leveraging recent advances in machine learning. There are two main heuristics used within AMG, namely selecting the coarse nodes for a given matrix (or graph) and construction of the interpolation operator between coarse and fine graphs. Recently, there have been studies on developing machine learning techniques to construct the interpolation operator [13, 14]. However, these methods require prior knowledge of the fine-coarse partitioning and the interpolation sparsity pattern. Hence, developing a learnable coarse-grid selection method is a necessary step in the development of fully learnable AMG algorithms. While much heuristic work has been done on the graph coarsening problem in this context (e.g., [15, 16]), optimal partitioning into coarse and fine nodes is known to be NP hard [17]. Nevertheless, the ML-based coarsening scheme that we propose yields a strictly better coarsening for a class of AMG solvers on planar grids compared to previous algorithms. We expand on this notion in Appendix D, where we note the limitations of comparing directly to the observed convergence of AMG algorithms based only on heuristics.

As we review in Section 3.1, the graph of $A$ in (1) can be partitioned into coarse and fine (a.k.a. fine-only) nodes; this has previously been presented as an optimization problem [17] and accompanied with a greedy algorithm for approximating the solution. However, for computational feasibility, this approach is rather simple and is known to break down in some cases [18]. An alternative is to use combinatorial optimization approaches such as simulated annealing [18]. While this produces good coarsenings, it is infeasibly expensive for large graphs. In this study, we develop a reinforcement learning method to achieve scalable (linear order in graph size) and high-quality fine-coarse partitioning, focusing on discretizations of the 2D Poisson equation,

$$-\Delta \phi = f, \tag{2}$$

where $\Delta$ is the Laplace operator and $f(x, y)$ and $\phi(x, y)$ are real-valued functions.

To the best of our knowledge, this work is the first to approach AMG coarsening using machine learning. Based on the optimization formulation of the coarsening problem introduced in [17] and outlined in more detail in Appendix D, we train a dueling double-DQN agent with topology adaptive convolution layers (TAGCN) [19] to yield a more efficient coarsening, as compared to previous studies. This optimization problem is tightly coupled, so that selecting one node to be in the coarse problem affects the eligibility of its neighbouring nodes for selection. Therefore, the process of coarsening is temporal in nature, and we model it as an RL problem, as discussed in Section 4. By using the optimization framework [17] as our starting point, we are able to prove that our RL method produces a convergent multigrid solver, and by combining evaluation with a graph decomposition algorithm [20, 21] we are able to coarsen efficiently.

The main contribution of this paper is to propose an RL method to train a grid-coarsening agent for the 2D Poisson problem which: (1) can coarsen arbitrary unstructured planar grids, including those much larger than the training grids, assuming bounded node degree, (2) can coarsen a grid with computational expense that is linear in the grid size (Theorem 2), (3) is guaranteed to produce a convergent multigrid method (Theorem 3), and (4) produces superior coarse grids to existing heuristic methods (Section 5).

## 2   Related work

**Machine learning for multigrid.**   While there is a long history of applying machine learning to multigrid algorithms [22], the last two years have seen a particular focus on using neural networks to generate multigrid interpolation operators. Greenfield et al. [14] used a residual architecture and unsupervised learning to train a fully-connected neural network that could outperform conventional

methods in generating the interpolation operator for diffusion problems on structured grids. In a later study [13], the model was further improved by employing graph neural networks and extending to positive semi-definite $A$ matrices. Given the coarse-fine splitting and the interpolation sparsity pattern, these neural networks are able to generate an interpolation operator which outperforms classical AMG methods in convergence factor. In other work, Schmitt et al. [23] used evolutionary computation methods to optimize a GMG algorithm. By learning from supervised data and using a U-Net architecture, Hsieh et al. [24] trained a convolutional network to improve a GMG algorithm for the structured Poisson problem. Katrutsa et al. [25] formulated the two-level V-cyle problem as a deep neural network and, using this formulation, they optimized the interpolation operator for GMG and evaluated the method on 1D structured grids.

**Reinforcement learning.** Reinforcement learning (RL) algorithms have demonstrated success in challenging decision-making problems such as complex games [26] and robotics [27]. RL algorithms largely fall into two categories: *value learning* and *policy optimization*. Value learning algorithms such as Deep Q-Learning [28] attempt to learn the so-called *Q function*, a function that guides an agent to select the best action in a given state. In contrast, policy optimization algorithms such as Proximal Policy Optimization (PPO) [29] try to maximize the possibility of actions that lead to higher rewards. In this paper, we use Dueling Double DQN [30, 31].

**Graph neural networks.** The natural modeling of many problems as graphs where local structure is important has led to the development of graph convolutional neural networks (GCNNs). GCNNS are commonly categorized into two main types: spectral and spatial [32]. First introduced by Bruna et al. [33], spectral methods define graph convolution as diagonal operators in the graph Laplacian eigenbasis. These methods, however, are highly domain specific and, once trained, they cannot adapt to different graph structures. Moreover, due to the Laplacian matrix eigendecomposition, they are computationally expensive. However, a number of recent studies [34, 35] have investigated how these limitations can be alleviated. Spatial methods, on the other hand, define graph convolution by locally propagating information. Message Passing Neural Networks (MPNNs) are a popular framework for spatial GCNNs [36]. A more general message passing framework was introduced by Battaglia et al. [37]. They introduced graph network blocks for learning relational data in the graph structure. Du et al. [19] introduced TAGCN, a GNN that is defined in the vertex domain of the graph. They have shown that the topologies of the learnable filters in TAGCN are adaptive to the topology of the graph on which convolution is performed. In this paper, we use a TAGCN for our RL agent.

## 3 Multigrid background

AMG algorithms infer a "grid" structure by using the graph of $A$ in (1). In the *setup* phase, a coarser grid with $n_c < n$ nodes is constructed, by selecting a subset of the nodes of the current grid. A full-rank interpolation operator, $P \in \mathbb{R}^{n \times n_c}$, is created to map vectors from the coarse grid to the fine grid, and a coarse-grid operator, $A_c \in \mathbb{R}^{n_c \times n_c}$, is also created, often using the Galerkin product, $A_c = P^T A P$. After setup, the AMG *solve* phase is executed by taking an initial guess for the solution, $x^{(0)} \in \mathbb{R}^n$, and performing a number of pre-relaxation (iterative linear solver) sweeps on the finest grid, often with Gauss-Seidel or weighted Jacobi, to damp high-energy errors in $x^{(0)}$. Then, the residual is restricted to the coarse grid, and the error equation, in the form of (1), is solved, and the solution is interpolation and added to the fine-level solution. Finally, a number of post-relaxation sweeps are performed to the corrected approximation. The AMG solve phase, which is analogous to that of geometric multigrid (GMG) [38], is outlined in Algorithm 1. Note that while we give the two-grid cycle only, the solution of the system at Step 5 can also be done approximately, recursing with the same algorithm to an arbitrary number of levels.

Since the AMG solution phase is an iterative algorithm, we must assess both the cost per cycle (measured by its *grid complexity*) and the convergence factor (measured by the reduction in the residual in each iteration) in order to quantify efficiency. Of these, convergence is more thoroughly studied. From a theoretical perspective, typical convergence results rely on making assumptions on the coarsening and construction of the interpolation operator [7, 8, 9, 10, 11] to bound the per-cycle convergence factor. Alternately, heuristic or other techniques (including ML approaches [13, 14]) are used to generate coarsenings and interpolation operators that may lead to good convergence. In contrast, grid complexity, defined as the sum of the grid sizes over all levels in the hierarchy divided by the size of the finest grid, measures the computational complexity of a cycle of an AMG

**Algorithm 1** Two-Level AMG Algorithm

---

1: **Input**: Sparse matrix $A \in \mathbb{R}^{n \times n}$, right-hand side $b \in \mathbb{R}^n$, initial guess $x^{(0)} \in \mathbb{R}^n$, interpolation matrix $P \in \mathbb{R}^{n \times n_c}$, coarse-grid matrix $A_c \in \mathbb{R}^{n_c \times n_c}$, convergence tolerance $\delta$, numbers of relaxation sweeps $N_1, N_2 \in \mathbb{N}$, and $k = 0$.
2: **repeat:**
3:      Perform $N_1$ pre-relaxation sweeps on $x^{(k)}$ to obtain $\tilde{x}^{(k)}$
4:      Calculate the residual: $\tilde{r}^{(k)} = b - A\tilde{x}^{(k)}$
5:      Project the residual to the coarse grid and solve: $A_c e_c^{(k)} = P^T \tilde{r}^{(k)}$.
6:      Interpolate and add the coarse-grid correction: $\hat{x}^{(k)} = \tilde{x}^{(k)} + P e_c^{(k)}$
7:      Perform $N_2$ post-relaxation sweeps on $\hat{x}^{(k)}$ to get $x^{(k+1)}$
8:      $k = k + 1$, compute $r^{(k+1)} = b - A x^{(k+1)}$
9: **until:** $||r^{(k+1)}|| < \delta$

---

algorithm, assuming that the dominant costs in the algorithm are proportional to the number of degrees of freedom on each level. Notably, however, a common theme in the theoretical approaches mentioned above [9, 10, 11] is that it is difficult to guarantee a fixed bound on the grid complexity while simultaneously bounding the resulting AMG convergence factor.

### 3.1 Grid coarsening and greedy coarsening

Classical AMG coarsening is based on heuristics to generate maximal independent subsets of the graph on each level [2]. While often effective, there are no guarantees on either the grid complexities of the generated hierarchies or the convergence properties of the resulting algorithm. In an attempt to address this, MacLachlan and Saad [17] proposed an optimization form of the coarsening problem, building on the work of MacLachlan et al. [9], which provides a rigorous bound on the two-level AMG convergence factor depending on properties of the grid partitioning. Define a binary variable indicating if a node is fine or coarse:

$$f_i = \begin{cases} 1 & \text{if } i \in F, \\ 0 & \text{if } i \in C, \end{cases} \tag{3}$$

where $F$ and $C$ denote the sets of fine and coarse nodes, respectively. The key element in the theory of [9] is suitable diagonal dominance of the submatrix of $A$ restricted to the $F$ set. This is encapsulated by the following optimization problem, which defines the largest $F$ set that satisfies the dominance criterion:

$$\text{maximize} \sum_{i=1}^{n} f_i \tag{4a}$$

$$\text{subject to } |a_{ii}| \geq \theta \sum_{j \in F} |a_{ij}| \text{ for all } i \text{ such that } f_i = 1. \tag{4b}$$

Here, $\theta$ is a dominance parameter that ties directly to the convergence theory in [9] (we use $\theta = 0.56$). Thus, solving (4) offers the lowest (two-grid) complexity to achieve a solver with a certain guaranteed convergence factor per iteration. In [17], MacLachlan and Saad proved that this optimization problem is NP-hard and proposed a greedy algorithm for approximating its solution. In [18], it is shown that the greedy algorithm is ineffective in solving (4) in some cases, and a simulated annealing approach (SA) is used instead. While that approach is effective, it has a high cost, due to the inefficiency of SA. Here, we aim to apply tools from ML to retain the effectiveness of the SA approach, but at a lower cost. Our proposed method is linear in problem size, while SA is less efficient in comparison.

## 4 Reinforcement learning for multigrid coarsening

While grid coarsening could be thought of as a function that maps a fine grid to a coarse grid, it is natural to formulate it as an iterative process that selects coarse nodes one by one, so that each decision can take into account the locations of previously selected coarse nodes. We thus start by reformulating the optimization problem (4) as a reinforcement learning problem.

## 4.1 Environment

To define the reinforcement learning environment, we start from the symmetric sparse linear system (1) with an $n \times n$ matrix $A$. We define a graph $G = (V, E)$ that has a node $i \in V$ for each variable index $i = 1, \ldots, n$ and an edge $(i, j) \in E$ if $A_{ij} \neq 0$ for $i \neq j$. Consider two binary variables $f_i \in \{0, 1\}$ and $v_i \in \{0, 1\}$ for each node $i \in V$. Here $f_i$ indicates whether node $i$ is a fine node, as in (3), and $v_i$ indicates whether node $i$ violates the diagonal dominance constraint (4b). Recall that $F = \{i \mid f_i = 1\}$ is the set of fine nodes and $C = \{i \mid f_i = 0\}$ is the set of coarse nodes. The **state space** is now defined by $\mathcal{S} = \{(f_i, v_i) \mid i = 1, \ldots, n\}$ which is $2n$ binary variables, two for each node. The **initial state** $s_0$ consists of all fine nodes, so $f_i = 1$ for all $i$, and $v_i$ is determined by:

$$v_i = \begin{cases} 1 & \text{if } f_i = 1 \text{ and } |a_{ii}| < \theta \sum_{j \in F} |a_{ij}|, \\ 0 & \text{otherwise.} \end{cases} \tag{5}$$

At each time step the agent chooses one violating fine node to convert into a coarse node. The **action space** is thus $\mathcal{A} = \{i \mid v_i = 1\}$. Given an action $a$ chosen in state $s$, the **next state** $s'$ changes $f'_a = 0$, so node $a$ becomes coarse, and $v'_i$ is recomputed for all nodes by (5). Note that $v'_i = v_i$ for all nodes that are not adjacent to the new coarse node $a$. The **reward** $r(s) = -|C|$ is the negative of the number of coarse nodes and the environment **terminates** when there are no more actions to take (that is, when no nodes are violating).

**Theorem 1.** *An optimal agent for the environment described above will exactly solve the optimization problem (4).*

*Proof.* Because each time step of the environment adds exactly one new coarse node, the total payoff function is $\rho = -n_c(n_c + 1)/2$, where $n_c$ is the number of coarse nodes at termination. The environment terminates at the first time that the constraint (4b) is satisfied and thus an optimal agent will minimize $n_c$ subject to (4b), therefore minimizing the number of steps. Equivalently, the number of fine nodes will be maximized, which is (4a). □

## 4.2 Agent and training

We want to learn an agent that is able to coarsen graphs for many different $Ax = b$ systems, which will be of different sizes and different graph topologies. We thus use a graph neural network agent that can adapt to arbitrary graph structures and a dueling architecture [31] to enable graph-size-invariant advantage estimation. Specifically, we use 3 TAGConv [19] layers for the agent, each consisting of 4-size filters and 32 hidden units. This means that the agent output at each node depends on information from nodes up to 12 hops away on the graph (3 layers $\times$ 4-size). We evaluate the agent on the graph $G$ associated with the matrix $A$ from (1), using $f_i$ and $v_i$ as node features and the elements of the $A$ matrix as edge features. TAGConv networks perform graph convolutions in the vertex domain and learn filters that are adaptive to the topology of the graph. They inherit the convolution properties of CNNs on rectangular grids but perform graph convolutions in the sense of graph signal processing.

The agent has two separate output heads in a dueling architecture [31], one to estimate the state value $V(s)$ and the other to estimate the state-action advantage $A(s, a)$. (In this paragraph, we temporarily overload the notation $A$ to refer to the advantage function rather than the matrix from (1), and $V$ to refer to the state value and not the vertex set of $G$.) These two output heads each consist of one TAGConv layer and the lower two TAGConv layers are shared, with a final averaging layer for the $V$ head. The state-action value function $Q(s, a)$ is, thus, the sum of the outputs from the two heads. While dueling architectures have benefits for training, they are also valuable for graph problems where the $Q$ and $V$ values are dependent on the size of the graph but $A$ is not, as is the case for multigrid coarsening. This size dependence arises because $Q$ and $V$ are estimating the number of coarse nodes needed for the graph, which scales with the graph size. In contrast, the advantage $A$ is only estimating the differential impact of coarsening a particular node, which will be a small number that depends only on the local graph structure and does not increase with the global graph size. Using a dueling architecture, we can train on small graphs, where we will learn $V$ and $A$ accurately for a certain size of graph, and then evaluate the agent on large graphs where $V$ will no longer be accurate but $A$ will continue to be correct. Since we only need the output of $A$ for evaluation, this provides a scale-invariant solution to the reinforcement learning problem.

**Algorithm 2** Evaluation Algorithm

---

1: Use Lloyd aggregation to decompose the node set into subdomains $\{V_1, V_2, ..., V_K\}$
2: **while** Constraint (4b) is not satisfied **do**
3:     Evaluate the advantage TAGCN network to obtain the advantage $A_i$ for each node
4:     **for** $k = 1$ to $K$ such that $V_k$ contains at least one node with $v_i = 1$ **do**
5:         $i = \underset{j \in V_k, v_j = 1}{\operatorname{argmax}} A_j$
6:         Coarsen node $i$
7:     **end for**
8: **end while**

---

To train the agent, we use Dueling Double DQN [30, 31]. Each episode consists of a different random convex triangular mesh in 2D with approximately 30 to 120 nodes. Meshes are generated by selecting a number of uniformly random points in 2D, taking their convex hull, and meshing the interior using pygmsh [39] (see Figure 1 as an example). On each mesh, we compute the piecewise linear finite-element discretization of the Laplacian, $A$, corresponding to the PDE (2), and perform a rollout of the environment in Section 4.1. All training and testing problems used a dominance factor of $\theta = 0.56$. We train for 10k episodes of DDQN with learning rate of $10^{-3}$, replay buffer of size $5 \times 10^3$, and batch size of 32. At the beginning of the training, the exploration $\epsilon$ value in the DDQN algorithm is set to 1 and decayed at rate of $1.25 \times 10^{-4}$ to a minimum value of $10^{-2}$. The target network weights are updated after each 4 episodes, and the frozen network is replaced by the target network after every 10 learning steps. See Appendix B for a discussion of the choices made for the network and its architecture.

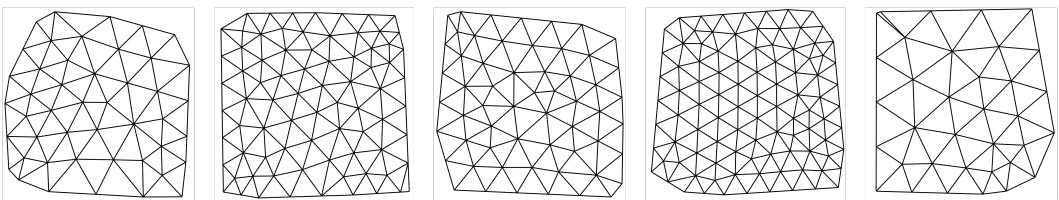

Figure 1: Examples training grids with sizes ranging from approximately 30 to 120 nodes.

### 4.3 Evaluation

To coarsen a graph, we can simply apply the trained agent to the graph, coarsen the fine node with the highest advantage output, and repeat. However, a direct implementation of this will incur a cost of $O(n^2)$ for a graph with $n$ nodes, because each step to coarsen a node requires an $O(n)$ cost to evaluate the TAGCN network and there are $O(n)$ coarse nodes required. Thus, we follow [18] and use a localization strategy to reduce cost.

To perform graph coarsening with $O(n)$ total cost, we combine graph decomposition via Lloyd aggregation [20, 21] with the trained agent. See Appendix C for a more detailed discussion of Lloyd aggregation. We first use the Lloyd aggregation algorithm to decompose the node set into a disjoint union of subdomains. Then, at each step, we evaluate the advantage TAGCN network on the entire graph and coarsen one node per subdomain (if there are any available). By decomposing into $O(n)$ subdomains of bounded size, this method terminates in $O(1)$ steps and results in an $O(n)$ cost for the entire coarsening procedure. Algorithm 2 details this method, Theorem 2 proves that it has $O(n)$ cost, and Figure 3 (right) shows numerical verification of the linear cost scaling. Furthermore, Theorem 3 uses the existing AMG theoretical results [17] to prove that Algorithm 2 necessarily results in a convergent multigrid method.

**Theorem 2.** *For grids with $n$ nodes having $n$-independent bounded node degree and $n$-independent bounded subdomain size when using Lloyd aggregation with a fixed number of cycles, the time complexity of Algorithm 2 is $O(n)$.*

*Proof.* Since the Lloyd subdomain size is bounded and the number of Lloyd aggregation cycles are fixed, generating the subdomain decomposition has $O(n)$ time complexity [20]. Evaluating each

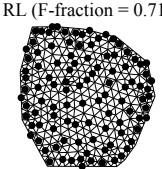
RL (F-fraction = 0.71)

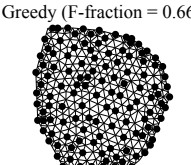
Greedy (F-fraction = 0.66)

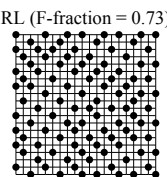
RL (F-fraction = 0.73)

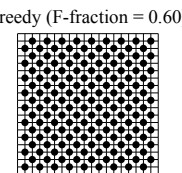
Greedy (F-fraction = 0.60)

Figure 2: Example coarsenings of meshes from the "Different Size" (left) and "Structured" (right) families of test grids, comparing RL and greedy coarsening algorithms for dominance factor $\theta = 0.56$.

TAGCN layer consists of $y = \sum_{\ell=1}^{L} G_\ell x_\ell + b\mathbf{1}_n$, where $L$ is the number of node features, $b$ is a learnable bias, $G_\ell \in \mathbb{R}^{n \times n}$ is the graph filter, and $x_\ell \in \mathbb{R}^n$ are the node features. Moreover, the graph filter is a polynomial in the adjacency matrix $M$ of the graph: $G_\ell = \sum_{j=0}^{J} g_{\ell,j} M^j$ where $J$ is a constant and $g_{\ell,j}$ are the filter polynomial coefficients. The bounded node degree of the graph implies that $M$ is sparse and the action of $M^j$ has $O(n)$ cost, and thus the full TAGCN evaluation also has $O(n)$ cost. Because the Lloyd aggregate size is bounded, we have $K = O(n)$ and so the for-loop in Algorithm 2 has $O(n)$ cost. Finally, the bounded Lloyd subdomain size also implies that the number of while-loop iterations in Algorithm 2 is independent of $n$, and thus the total cost is $O(n)$. $\qquad\square$

**Theorem 3.** *Algorithm 2 is guaranteed to terminate and satisfy constraint* (4b)*. Moreover, if matrix $A$ in linear system* (1) *is diagonally dominant, the resulting two-grid multigrid cycle (Algorithm 1) is guaranteed to converge with a convergence factor bounded independently of $n$.*

*Proof.* Since every iteration of Algorithm 2 coarsens at least one node, the algorithm must terminate since the graph is finite. Moreover, as long as there are fine nodes violating constraint (4b), the algorithm does not terminate, and so constraint (4b) must be satisfied at termination. Finally, from Theorems 3 and 5 in MacLachlan and Saad [17], diagonal dominance of $A$ and (4b) are sufficient to guarantee uniform convergence of the two-grid cycle. $\qquad\square$

## 5 Numerical Experiments

To test the performance of the RL method for multigrid coarsening, we use six different families of test grids, as summarized in Table 1. Each of these families explores a different type of grid structure and has a parameter to vary this structural element. For all tests, we used the agent trained following Section 4.2 and evaluated using Algorithm 2 with mean Lloyd aggregate size of 400 nodes and 1000 Lloyd cycles. The code [1] was implemented using PyTorch Geometric [40], PyAMG [41], and NetworkX [42]. All computation was performed using CPU-only on an 8-core i9 MacBook Pro. The training time was approximately 5 hours and the average evaluation time was approximately 20 seconds per test grid. Example grid coarsenings are shown in Figure 2.

**Evaluation metrics: F-fraction and effective convergence factor.**  The primary metric for coarse grid quality that we are optimizing is the number of fine nodes, which should be maximized as in (4a). To quantify this, we use the F-fraction (higher is better), which is the ratio of the number of fine nodes to the total number of nodes. As a secondary metric, we use the effective convergence factor (lower is better) of the two-level multigrid method formed from the coarse grid using the reduction-based AMG interpolation operator from [9, 17]. The effective convergence factor is defined as the per-cycle AMG convergence factor raised to the power of one over the AMG grid complexity, which is a measure of the convergence factor per unit of computational work.

**Comparison methods: Greedy and theoretical bounds.**  To assess the performance of our RL coarsening method, we compare with the simple greedy method of MacLachlan and Saad [17], which progressively coarsens the grid by greedily selecting the node with the lowest value of $|a_{ii}|/\sum_{j \in F} |a_{ij}|$ at each step (see (4b) for reference). While the greedy method provides a good

---

[1] All code and data for this paper is at `https://github.com/compdyn/rl_grid_coarsen` (MIT licensed).

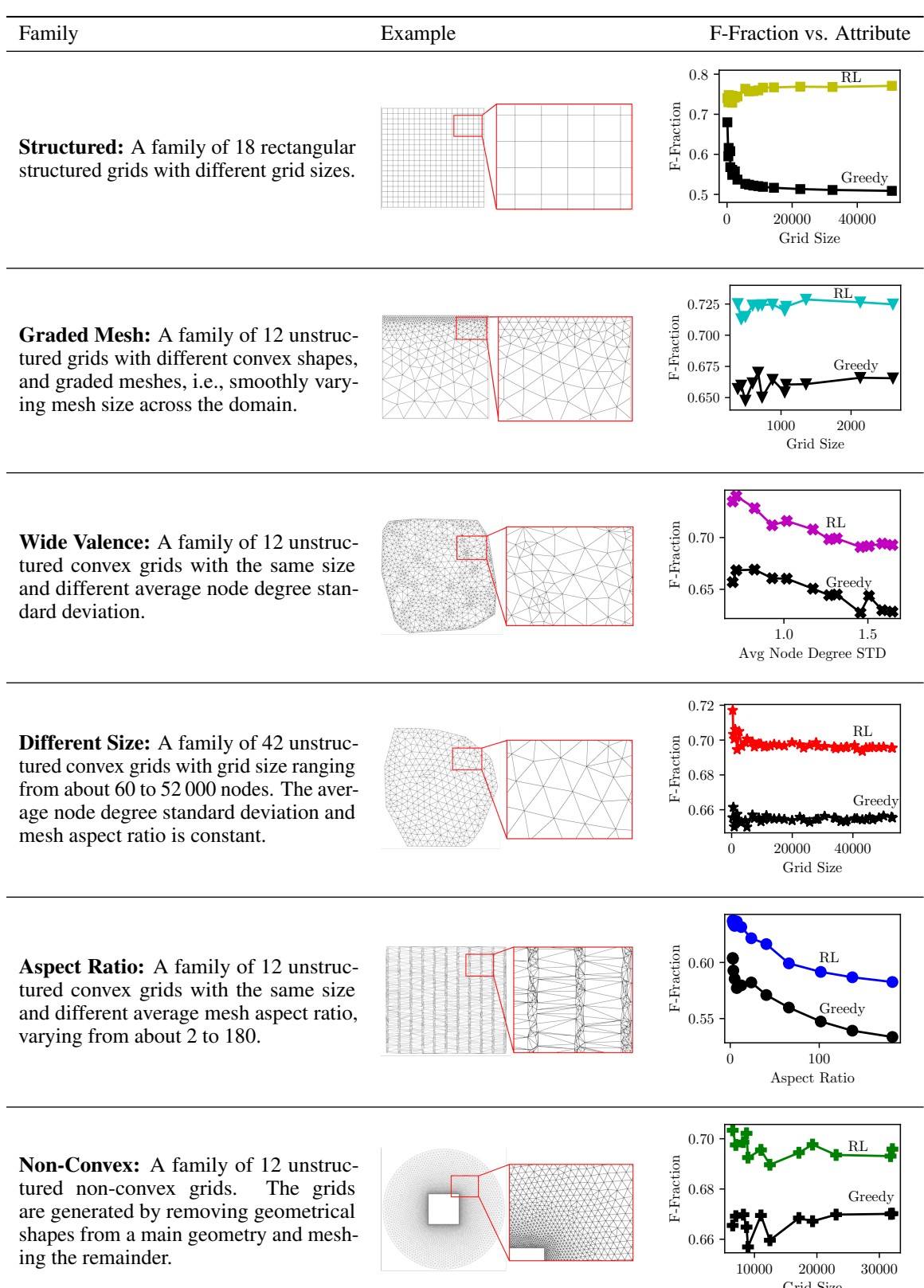

Table 1: Families of test grids used for numerical experiments, showing F-fractions (higher is better).

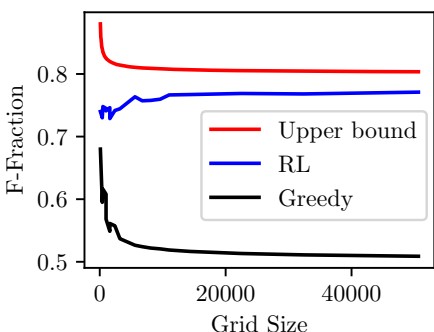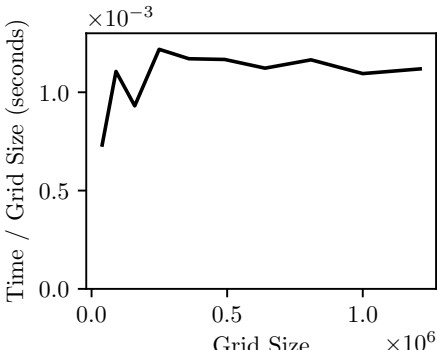

Figure 3: Left: F-fraction for the RL method and comparison methods (higher is better) for the Structured family of grids. Right: Evaluation time divided by grid size, showing linear scaling in grid size.

baseline, we can also derive a theoretical upper-bound for the structured grid family, as described below in Theorem 4.

**Theorem 4.** *For the Poisson problem (2) discretized with a 5-point finite difference stencil on a structured grid of size $n_x \times n_y$, the constraint (4b) implies that the F-fraction, $f$, is bounded by:*

$$f \le 0.8 + \frac{2}{n_x} + \frac{2}{n_y} - \frac{4}{n_x n_y}. \tag{6}$$

*Proof.* To satisfy constraint (4b) for the 5-point stencil, every pentomino (a five cell structure obtained after removing corners of a $3 \times 3$ grid) within the grid must have at least one coarse node. Given an $n_x \times n_y$ rectangular grid, there are at least $(n_x - 2)(n_y - 2)$ pentominoes entirely lying within the grid, so there are at least this many coarse nodes. On the other hand, every coarse node appears in at most five different pentominoes. Letting $n_c$ be the number of coarse nodes, this implies $(n_x - 2)(n_y - 2) \le 5n_c$. Rearranging this inequality and using $f = 1 - n_c/(n_x n_y)$ gives (6). □

**Structured grid coarsening results.**  The left panel in Figure 3 shows the performance of the RL method compared to the greedy baseline and the theoretical upper bound. For large grids, the RL method achieves 96.4% of the theoretical upper bound, which is 51.5% higher than the greedy method. Most significantly, RL performs better on all grid sizes, with improving performance as the grid size increases. The right panel in Figure 3 confirms that the time to coarsen is linear in the grid size, for structured grids with up to 1.2 million nodes.

**Unstructured grid coarsening results.**  For the unstructured grid families, the F-fractions obtained by the RL and greedy methods are shown in the right-most column in Table 1. Here, we see that the RL method always substantially outperforms the greedy method for every grid family, and for all parameter variations. These families include very diverse and challenging grids, including those with very wide valence (some nodes with high degree and others with low degree) and those with high aspect ratio elements (up to 180). This data is summarized in the left panel of Figure 4. The right panel of this figure shows the effective convergence factors of the resulting two-level multigrid methods. The methods were not directly optimizing for this metric, but we nonetheless see that the RL and greedy grids have generally similar convergence factors. The main exceptions are for cases where the greedy method has dramatically lower F-fraction than the RL method, such as the structured grid family. Further details on these numerical results are given in Appendix A.

## 6   Conclusion

In this paper, we introduce a reinforcement learning (RL) method utilizing graph convolution neural networks (GCNN) to approach the NP-hard problem of grid coarsening for algebraic multigrid methods (AMG). We use our method for the 2D Poisson problem on a wide variety of unstructured

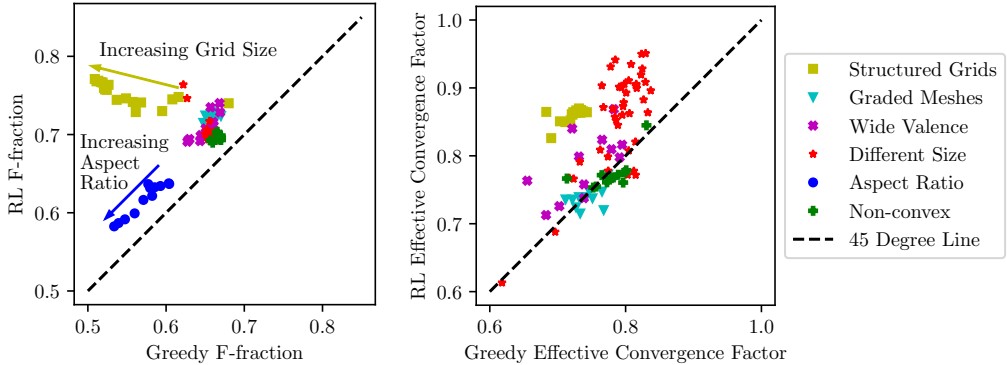

Figure 4: Comparison between RL and greedy algorithms on the F-fraction (higher is better) and effective convergence (lower is better) metrics, for all families of test grids.

grids. Moreover, we demonstrate that our approach strictly outperforms the existing heuristic method [17], while still preserving the theoretical guarantees for convergence of the resulting multigrid solver. Our RL agent is trained on small grids and uses a dueling advantage and value function decomposition to apply to arbitrarily large grids without performance degradation. Furthermore, we prove that our graph-decomposition-based evaluation algorithm scales linearly with grid size. The main limitations of the current work is that the agent was specialized to a single PDE (Poisson's equation) on 2D grids and it is unclear how readily it will generalize to other problems. In future work, we propose to extend the approach considered here to more general families of AMG algorithms (and, indeed, to other problems and algorithms in numerical linear algebra), aiming to achieve improved efficiency in the resulting AMG solvers.

## Funding Transparency Statement

Funding in direct support of this work: NSERC grant RGPIN- 2019-05692 to SM. The authors have no competing interests to declare.

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
