# Optimization-Based Algebraic Multigrid Coarsening Using Reinforcement Learning

**Ali Taghibakhshi**
Mechanical Science and Engineering
University of Illinois at Urbana-Champaign
Urbana, IL 61801, USA
alit2@illinois.edu

**Scott MacLachlan**
Mathematics and Statistics
Memorial University of Newfoundland
and Labrador
St. John's, NL, Canada
smaclachlan@mun.ca

**Luke Olson**
Computer Science
University of Illinois at Urbana-Champaign
Urbana, IL 61801, USA
lukeo@illinois.edu

**Matthew West**
Mechanical Science and Engineering
University of Illinois at Urbana-Champaign
Urbana, IL 61801, USA
mwest@illinois.edu

In this document, we provide a convergence plot for all the families with guaranteed convergence A, a discussion about the choice of the graph neural network and architecture B, a more detailed discussion about the clustering algorithm we used for our method (Lloyd aggregation C), and the theoretical guarantees for the AMG coarsening D.

## Appendix A   Convergence of AMG

Figure 1 shows the convergence factor for all of the families (except for the Aspect Ratio family, for which the theoretical convergence factor bound does not apply) and the convergence factor for the different size family against grid size. It is noteworthy that, in this paper, we have directly optimized the size of the coarse grid, which sacrifices convergence in exchange for achieving smaller coarse grids, justified by the fact that the guaranteed convergence factor is always 0.977, according to convergence theory in Appendix D.

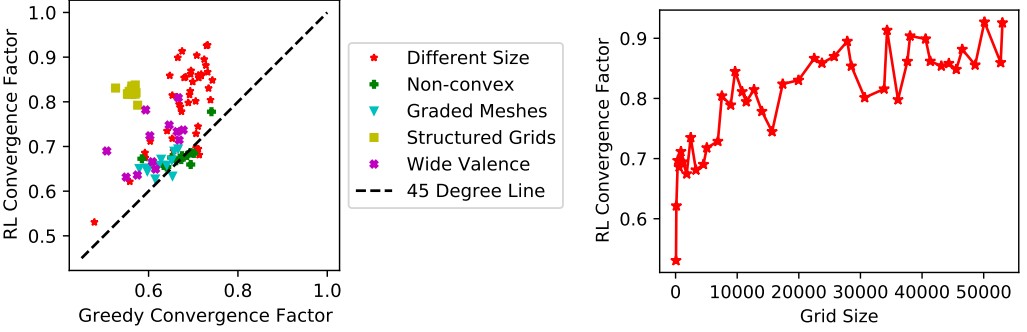

Figure 1: Left: Measured convergence factors for different families with guaranteed convergence. According to the theory in Appendix D, the guaranteed convergence factor is 0.977. Right: measured RL convergence rate for the Different Size family versus grid size.

35th Conference on Neural Information Processing Systems (NeurIPS 2021).

## Appendix B    Choice of Graph Neural Network and Architecture

We use TAGConv [1] layers as graph convolution in our architecture. This choice is motivated by the fact that TAGConv has linear time complexity in graph size (assuming a size-independent upper bound on node degree). Moreover, it is a generalized form of the conventional GCN [2], and it is scalable, i.e., we can train such a network on certain graph types, yet test them on larger graphs with different topologies. In this work, we obtain our experiment results using 3 TAGConv layers for the agent, each consisting of 4-size filters with 32 hidden units. This architecture results in the neural network output at each node being dependent only on information from nodes up to 12 hops away from it (3 layers × 4-size). For comparison, we trained another agent, using the same RL algorithm (i.e. Dueling Double DQN [3]), but this time with 6 TAGConv [1] layers, each consisting of 8-size filters with 64 hidden units, leading to output of the network at each node being dependent on nodes within 48 hops away (6 layers × 8-size). Comparing the results of the two networks, we observe that adding more graph convolution layers does help the agent achieve better results on the training set; however, when testing on grids noticeably larger than those in the training set, the larger network does not outperform the smaller network. We conclude that the initial choice of the network parameters provides the agent with enough information about its local neighborhood to outperform the greedy algorithm of [4], while there is no substantive benefit to adding more complexity (and cost) to the network. A more thorough study of the network complexity and training time is left for future work.

## Appendix C    Lloyd Aggregation

Lloyd aggregation is a k-means-based clustering algorithm which is obtained directly from the general form of Lloyd's algorithm [5, 6]. Consider a 2D planar graph, $G$, the set of all edges $E$, the set of all of its nodes $V$, and $V_c \subseteq V$. The nodes in the $V_c$ set serve as the centers of the clusters. The regions are obtained based on the closest center to each graph node, where the distance is measured by the number of edges covered in the shortest path between two nodes (denote distance between node $i$ and $j$ by $d_{ij}$). Define the centroid of a region as the farthest node from the boundary, and break the possible ties by choosing one randomly. The modified Bellman-Ford algorithm is commonly used for obtaining the nearest center to every node in $V$ and its associated distance [5, 7]. Let $N$ be a list of graph nodes whose $i$-th element is the nearest center to the $i$-th node of the graph, and let $D$ be the list of its distances; then, the modified Bellman-Ford algorithm is shown in Algorithm 1.

---

**Algorithm 1** Modified Bellman-Ford

1: **Input** $E$: The set all edges, $V$: The set of all nodes, $V_c$: The set of initial center nodes.
2: $D(i) = \infty \; \forall_{i \in \{1,2,...,|V|\}}$
3: $N(i) = -1 \; \forall_{i \in \{1,2,...,|V|\}}$
4: **for** $c \in V_c$ **do**
5: $\quad$ $D(c) \leftarrow 0$
6: $\quad$ $N(c) \leftarrow c$
7: **end for**
8: **while** True **do**
9: $\quad$ Finished $\leftarrow$ True
10: $\quad$ **for** $(i, j) \in E$ **do**
11: $\quad\quad$ **if** $D(i) + d_{ij} < D(j)$ **then**
12: $\quad\quad\quad$ $D(j) \leftarrow D(i) + d_{ij}$
13: $\quad\quad\quad$ $N(j) \leftarrow N(i)$
14: $\quad\quad\quad$ Finished $\leftarrow$ False
15: $\quad\quad$ **end if**
16: $\quad$ **end for**
17: $\quad$ **if** Finished **then**
18: $\quad\quad$ **return** $D, N$
19: $\quad$ **end if**
20: **end while**

---

Once the $D$ and $N$ reach their correct values, Algorithm 1 terminates. Given a fixed number of center nodes, Lloyd's algorithm modifies the clusters in each iteration by selecting the centroid of every region as its new center, then using the modified Bellman-Ford algorithm to calculate new distances

and nearest centers. Given updated center positions, it reshapes the new clusters. The Full Lloyd algorithm is shown in Algorithm 2, where we define the set of border nodes, $B$, as the set of all nodes that are connected by an edge to a node that has a different nearest center node.

---

**Algorithm 2** Lloyd Aggregation

---
1: **Input** $K$: Number of iterations, $E$: The set of all edges, $V$: The set of all nodes, $V_c$: The set of initial center nodes.
2: **for** $i = 1, 2, 3, ..., K$ **do**
3:     $D, N \leftarrow$ Modified Bellman-Ford$(E, V, V_c)$
4:     $B \leftarrow \emptyset$
5:     **for** $(i, j) \in E$ **do**
6:         **if** $N(i) \neq N(j)$ **then**
7:             $B \leftarrow B \cup \{i, j\}$
8:         **end if**
9:     **end for**
10:     $D, X \leftarrow$ Modified Bellman-Ford$(E, V, B)$
11:     $V_c \leftarrow \{i \in V : D(i) > D(j) \ \forall_{N(i)=N(j)}\}$
12: **end for**
13: **return** $N$

---

**Time Complexity:** Assuming each node's initial distance to a center node is bounded independently of $|V|$, and also assuming that each node's degree is bounded independently of $|V|$, Algorithm 1 runs a $|V|$-independent number of iterations to determine one nearest center node for every point. So, we conclude that Algorithm 1 is $O(|V|)$ in our case. This is run a $|V|$-independent number of times in Algorithm 2 to get the clustering, resulting in an $O(|V|)$ clustering algorithm.

## Appendix D    Reduction-based AMG

Algebraic multigrid (AMG) methods are a family of iterative solution algorithms for linear systems of equations. First developed in the 1980's [8, 9], they have become a commonplace tool in the solution of discretized elliptic PDEs, particularly for applications such as the modeling of flow through porous media. As in geometric multigrid, AMG algorithms make use of complementary processes of *relaxation* and *coarse-grid correction* to reduce all modes of error in an approximate solution. In geometric multigrid methods, the typical approach is to fix the details of the coarse-grid correction process and develop relaxation schemes that provide sufficient error reduction over the complementary modes to ensure effective convergence overall. AMG methods take the opposite approach, fixing the details of the underlying relaxation scheme and constructing a coarse-grid correction process to complement it.

While commonly implemented recursively, we focus on *two-level* cycles in this work. In this setting, the fine-grid relaxation scheme is characterized by its error-propagation operator, where the current approximation to the solution of $Ax = b$, $\tilde{x}$, is updated by

$$\tilde{x} \leftarrow \tilde{x} + M(b - A\tilde{x}),$$

using some matrix $M$ — e.g., $M = \omega D^{-1}$ with some weight and the diagonal of $A$, in the case of weighted Jacobi. From this, the error $e = x - \tilde{x}$ is updated as $e \leftarrow (I - MA)e$. Typically, the coarse-grid correction process in AMG is specified by the choice of an interpolation operator, $P$, that maps from the (smaller) coarse space to the given fine space. Then, coarse-grid correction updates an approximation, $\tilde{x}$, as

$$\tilde{x} \leftarrow \tilde{x} + P(P^T A P)^{-1} P^T (b - A\tilde{x}),$$

and updates the error as $e \leftarrow (I - P(P^T A P)^{-1} P^T A)e$. The main distinction between different AMG approaches, then, lies in the construction of $P$. Classical AMG schemes construct $P$ by first determining its row dimension (the *coarsening* or *partitioning* phase), then fixing a sparsity pattern (to ensure suitable sparsity of $P^T A P$) and computing the nonzero entries.

While AMG methods are well-regarded as among the most efficient solvers available for the large, sparse, linear systems that come from the discretization of elliptic PDEs, the available theory governing their convergence often offers only poor prediction of their actual convergence rates

(see [10] for an overview). In particular, AMG algorithms are typically driven by heuristic (greedy) graph algorithms that make achieving guaranteed convergence rates difficult. In practice, there are only a few AMG frameworks that offer guarantees of even two-level convergence rates, and these are often limited to specific problems, such as the graph Laplacian [11, 12]. In this paper, we make use of the reduction-based AMG (AMGr) framework, introduced in [13], which offers guarantees of convergence for a somewhat wider class of problems, including finite-element discretizations of isotropic diffusion on some classes of mesh. We emphasize here that, while working in a theoretically justified setting is attractive, AMG algorithms with guaranteed convergence rates typically attain *worse* convergence than those based solely on heuristics. Thus, comparisons of the convergence of classical (heuristic-based) AMG algorithms directly with AMGr is potentially misleading, since it is a comparison of heuristics (often tuned for standard test problems) against theoretical guarantees.

A key advantage of the AMGr paradigm (aside from the convergence-rate guarantee) is that it offers a prescriptive construction of the AMG interpolation operator, $P$, once the partitioning is determined. In this area, it is typical to consider the problem in a permuted form, where we split the set of degrees of freedom (columns) and equations (rows) of $A$ into their "fine" and "coarse" subsets, writing $A = \begin{bmatrix} A_{ff} & -A_{fc} \\ -A_{fc}^T & A_{cc} \end{bmatrix}$, where symmetry is assumed and we introduce a sign flip for later convenience. The theoretical foundation of AMGr is the assumption that the submatrix, $A_{ff}$, can be written as $A_{ff} = D + \mathcal{E}$, where $D$ is a matrix that has a sparse inverse, while $\mathcal{E}$ is, in some sense, small. This is formalized in the main theorem of [13]. Here, we use the standard notation $A \geq 0$ to denote that matrix $A$ is positive semi-definite (that $y^T A y \geq 0$ for all vectors $y$).

**Theorem 1.** *Let $A = \begin{bmatrix} A_{ff} & -A_{fc} \\ -A_{fc}^T & A_{cc} \end{bmatrix} \geq 0$ be symmetric and let $A_{ff} = D + \mathcal{E}$ for symmetric matrix $D$, with $0 \leq \mathcal{E} \leq \epsilon D$ and $\begin{bmatrix} D & -A_{fc} \\ -A_{fc}^T & A_{cc} \end{bmatrix} \geq 0$. Define interpolation operator $P = \begin{bmatrix} D^{-1} A_{fc} \\ I \end{bmatrix}$ with coarse-grid operator $T = I - P(P^T A P)^{-1} P^T A$ and, for $\sigma = \frac{2}{2+\epsilon}$, define the relaxation operator to have error propagator $G = I - \sigma \begin{bmatrix} D^{-1} & 0 \\ 0 & 0 \end{bmatrix} A$. The two-grid AMGr error propagation operator defined by $TG$ satisfies:*

$$\|TG\|_A \leq \left( \frac{\epsilon}{1+\epsilon} \left( 1 + \frac{\epsilon}{(2+\epsilon)^2} \right) \right)^{\frac{1}{2}} < 1. \tag{1}$$

As written, this theorem provides a convergence guarantee, but neither a predictive algorithm for determining the partitioning nor for the choice of $D$ once $A_{ff}$ is known. These were later provided in [4], where it was realized that diagonal dominance of $A_{ff}$ is sufficient to provide a prescriptive choice of $D$.

**Theorem 2.** *Let $A$ be symmetric and positive definite, and let $\theta \in (\frac{1}{2}, 1]$ be given. If*

$$\theta \leq \theta_i = \frac{a_{ii}}{\sum_{j \in F} |a_{ij}|} \quad \forall i \in F,$$

*and $D$ is the diagonal matrix with entries $d_{ii} = (2 - \frac{1}{\theta_i}) a_{ii}$ for all $i \in F$, then $A_{ff} = D + \mathcal{E}$ with $0 \leq \mathcal{E} \leq \epsilon D$ for $\epsilon = \frac{2-2\theta}{2\theta-1}$.*

This result guarantees the first of the two conditions on $D$ in Theorem 1, but not the second, that $\begin{bmatrix} D & -A_{fc} \\ -A_{fc}^T & A_{cc} \end{bmatrix} \geq 0$. A further result in [4] addresses this.

**Theorem 3.** *Let $A$ be symmetric, positive definite, and diagonally dominant, and let $\theta \in (\frac{1}{2}, 1]$ be given. If*

$$\theta \leq \theta_i = \frac{a_{ii}}{\sum_{j \in F} |a_{ij}|} \quad \forall i \in F,$$

*and $D$ is the diagonal matrix with entries $d_{ii} = (2 - \frac{1}{\theta_i}) a_{ii}$ for all $i \in F$, then $\begin{bmatrix} D & -A_{fc} \\ -A_{fc}^T & A_{cc} \end{bmatrix} \geq 0$.*

Motivated by these results, the optimization problem in Section 3.1 of the main paper was proposed in [4] to determine the partitioning, by maximizing the size of $A_{ff}$ subject to ensuring the condition on diagonal dominance is met. This problem is shown to be NP-hard, motivating a greedy algorithm for its solution. It is that greedy algorithm that we compare against in Section 5 of the main paper. We note that the matrices arising from the aspect ratio family of problems are generally not diagonally dominant; thus, although they are an interesting and challenging test set, they do not fall into the class of problems for which convergence guarantees are given by the results above. For this reason, we do not include convergence results for this class.