# OpenReview forum: "Optimization-Based Algebraic Multigrid Coarsening Using Reinforcement Learning"
_NeurIPS.cc/2021/Conference — NeurIPS 2021 Poster_

### Official Review · Reviewer_nCB6 · 2021-06-26

**Rating:** 3
**Confidence:** 4

**Summary:**

The authors in the article have made a contribution to Algebraic Multigrid Coarsening. The AMG method is used to solve sparse linear systems. They proposed to replace heuristic approach used in the coarsening process by an Reinforcement Learning scheme. They have used Duel Double DQN with TAGCN to implement the RL based scheme. The paper describes in detail the problem formulation in terms of RL and presents results of the proposed algorithms. There are some important issues of concern regarding the submission.

**Main Review:**

 - The presented algorithm is only tested on 2D Poisson problem and they admit that they do not know how the algorithm will behave on other systems. This is a big hurdle in the acceptance of the paper. The contributions in the abstract and in section 1 are not formulated to support the view described earlier. A normal reader is prone to believe that the algorithm "...can coarsen arbitrary unstructured grids, including those much larger than the training grid" or is able to "...learn to perform graph coarsening on small training graphs and then be applied to unstructured large graphs"
- It is assumed that the structural bias learned by RL during the training will not allow it to be used in different scenarios. So the algorithm may have shown good result for one set of problems, but it may need to be trained from scratch for any other type of problem.
- Although, the authors claim that the algorithm is faster than any other technique, but there is no performance comparison present in the article.
- The code provided by the authors is not useful. It is hard to know the dependencies and even after that effort one comes to know that there are missing files in the code.
- For example, this line produces error in "from DuelingNet2 import Net as Net_MPNN" in MG_Agent.py
- I do not agree with the authors that the complexity of Algorithm 2 is O(n). The Complexity of Algorithm 2 depends on the structure of matrix A. Finding arg max could itself have a complexity of O(n) in normal circumstances. So the whole algorithm may have the complexity of O(n^2).
- It is not mentioned how the accuracy of the result is affected after grid coarsening. More coarse graph may also lead to bad results.

**Time Spent Reviewing:**

5

---

> ### Author Response · Authors · 2021-08-10
> **Addressing and amending the misunderstandings; notes on generalizability; fixed repository issues.**
>
> 1- We only intended to make claims about solving the same finite-element discretization of the same PDE (2D Poisson) on graphs which have a similar structure to our training graphs, which means planar graphs. To avoid misleading the reader, we will add “planar” and “assuming bounded node degree” to the abstract and introduction.  We agree that there is a large research gap between the limited question that we answer and the broader question that we would like to answer about more general systems of equations.
>
> 2- We agree that, as we have proposed, the method would require retraining for other PDEs. Obtaining methods which can generalize over multiple PDEs or graph embedding dimensions is an important problem that we hope can be addressed in future work by us and other groups. We have further clarified this in the abstract and introduction.
>
> 3- We apologize for the confusion. We have changed “fast” to “scalable (linear order in graph size)” in the revised manuscript.
>
> 4- We apologize for the errors in the provided code. The GitHub repository has been updated, and we believe that the software should now run on any system that satisfies the dependency requirements.
>
> 5- We agree with the reviewer that, in general, the algorithmic cost of the argmax would be O(n). In this case, however, each argmax is taken over a subdomain which is assumed to be of bounded size, and so each argmax has O(1) cost. We have changed the proof of Theorem 2 to clarify why the cost of the argmax in this setting is O(1).
>
> 6- We are not sure that we understand the reviewer’s comment and we would appreciate the clarification. We will add new theoretical results to the appendix to show that our full AMG method is guaranteed to converge to the exact answer, due to the fact that it always satisfies the constraint in Equation 4(b).

---

> > ### Comment · Reviewer_nCB6 · 2021-08-18
> > **Further elaboration of point 6**
> >
> > Any continuous system suffers loss of information due to discretization. I just wanted to see case study in which AMG method is used to solve the system, then a comparison is made how much numerical error was induced by the current state of the art algorithm and how much numerical error was induced by your algorithm.

---

> > > ### Author Response · Authors · 2021-08-23
> > > **Discretization error**
> > >
> > > We thank the reviewer for their clarification.  We agree that any discretization method introduces an error between the discrete solution and the solution of the continuous system.  Further error is introduced when, as in this case, error is made in the solution of the discretized system.
> > > If $u_c$ is the solution of the continuous system, $u_d$ is the solution of the discrete system, and $u_a$ is the approximate solution to the discrete system generated by our AMG method, a straightforward calculation (using the triangle inequality) gives:
> > > $\| u_c - u_a \| \leq \| u_c - u_d \| + \|u_d - u_a \|.$
> > > The discretization methodology should control the first term on the right-hand side.  Here, we use standard piecewise linear finite elements, as are appropriate for many practical problems in scientific computing, leading to well-known bounds on this term.  Robust error estimators can, in practice, be used to estimate or bound the second term, which is commonly referred to as the iteration error (in contrast to the first term, the discretization error).  In our original submission, we did not include metrics such as time-to-solution, so the selection of a stopping criterion to control the iteration error was not a relevant parameter.  We now propose to include such experiments in our revised manuscript, and will choose a stopping tolerance for the iterations in order to bound the total error, $\|u_c - u_a\|$, by a small factor times the discretization error, $\|u_c - u_d\|$.  This ensures that our algorithm introduces only a small additional error to the underlying discretization, which would be comparable with that generated by any solution algorithm for the same discretization.

---

### Official Review · Reviewer_oM5M · 2021-07-13

**Rating:** 6
**Confidence:** 4

**Summary:**

An AMG coarsening algorithm is proposed based on reinforcement learning on graph neural networks. The coarsening is determined from the diagonal dominance criterium for AMG as proposed by MacLachlan and Saad. Numerical experiments on a wide variety of 2D meshes (structured and unstructured) for the Poisson problem show excellent performance compared to the simple greedy algorithm proposed by MacLachlan and Saad. Theoretical results are fairly limited but the approach is very promising algorithmically. However, more meshes (3D), elliptic operators (smooth and discontinuous coefficients), and better comparison to the state-of-the art should have been tried to truly assess the novelty and viability of the proposed algorithm.

**Main Review:**

Questions / remarks:
1) Can the proposed Reinforcement Learning (RL) solution be used as starting value for the NP hard problem (4) when solved as an integer programming? This would also allow to deduce how far from optimality the RL solution is.

2) How is an optimal agent defined in Thm 1? As minimising the number of actions? It is worth point out that in practice one does not find an optimal agent.

3) Sec 4.2 - line 204: Please give more details (in the appendix) about the "random convex triangular mesh" and its FEM discretisation. How is the mesh generated? Which FEM discretisation (order of elements)?

4) Please explain briefly (in the appendix) how Lloyd aggregation works and why it would result in an $O(n)$ cost.

5) Fig 3: Add other metrics like AMG grid complexity and convergence factor. Effective convergence factor combines, which is nice, but could be difficult to interpret.

6) Fig 2: add the type of mesh in the legend.

7) While I find the results of Poisson on unstructured meshes very impressive, more experiments should be done truly assess the novelty and viability of the proposed ideas:
 - 3D (un)structured
 - Variable coefficients (smooth and discontinuous) elliptic on 2D/3D
 - Helmholtz problems
 - Compare to state-of-the-art and not only to the greedy approach: standard AMG with heuristic coarsening and other ML optimized MG.
 - Maybe not needed but nice: actually solve the PDE and compare time, cost.

It is probably not feasible to perform all these tests, but I find doing at least a few of them necessary to make the submission acceptable for Neurips. It is recommended to put much more experiments in the appendix.

**Time Spent Reviewing:**

4h

---

> ### Author Response · Authors · 2021-08-10
> **Addressing all the reviewer's comments; adding details for Lloyd algorithm and its O(n) time complexity; further studies on the problem, and more details in the appendix.**
>
> 1- This is a very interesting idea, and we would love to return to this in future work.
>
> 2- The optimal agent minimizes the expected sum of rewards; according to our problem formulation, an optimal agent will minimize the number of actions it needs to take until termination. We will point this out in the revised manuscript for better clarity.
>
> 3- Meshes are generated by selecting a number of uniformly random points in 2D, taking their convex hull, and meshing the interior using pygmsh. The FE discretization is piecewise linear. We will mention these points in the relevant section when we revise the manuscript.
>
> 4- In the appendix, we will provide the full Lloyd aggregation algorithm and will show that, under the assumptions of Theorem 2, it has O(n) time complexity.
>
> 5- We will add these plots in the appendix. We note that, for 2-level methods, grid complexity is a simple transformation of the F-fraction (it is 2 minus the F-fraction). We will mention this along with adding the plots.
>
> 6- We will make sure to add the mesh type in the caption.
>
> 7- All the points mentioned by the reviewer are great questions, and we will address them in future work. A key point about this work is that we choose to consider an AMG formulation that sacrifices observed performance for guaranteed mesh-independent convergence bounds.  In practice, the convergence bounds achieved with such theoretical guarantees are worse than those that can be achieved for these problems using the standard heuristics that are part of state-of-the-art AMG.  Bridging this gap is an important question in theoretical computer science that we are keen to address in future work.  We will add text to the manuscript to clarify this point.  Additionally, we will add a variable-coefficient example, and details on the comparison of time/cost for solving PDEs using the two AMG coarsening approaches.  In future work, we hope to tackle the question of developing and validating an end-to-end RL approach, including using solver performance as the RL reward.

---

### Official Review · Reviewer_hVW8 · 2021-07-15

**Rating:** 7
**Confidence:** 4

**Summary:**

The authors propose using RL agent via a TAGCOnv network in DQN to select coarse points in the procedure of coarsening algebraic grids. Theoretical results are established for the integration of the RL into a convergent scheme, and the complexity of the algorithm is at O(n) via a Lloyd based aggregation to constrain the coarsening to an O(n) selection.

**Main Review:**

Overall the paper clearly proposes a useful improvement of the AMG algorithm. The novelty here is in combining RL other AMG technique which Im not familiar with in prior work. The authors give sufficient empirical validation to the advantage of the suggested combination. The results are clearly of interest to practitioners who use AMG solvers in a variety of settings from solving pdes to other general systems in 2D grids. The authros clearly (and honestly) state that the generalization to other problems is still unknown.
There are still some details missing that I would be grateful if the authors could address in the rebuttal:
1.	I assume that the Lloyd based aggregation is a k-means essentially, hence the question is how to determine K so that the partition will still be O(n) but the size of each cluster will be approximately similar to that of the other clusters to allow faithful sampling from each.
2.	In this context, For example, I would expect the family of Aspect ratio to be fairly challenging. For some reason (could be a typo) the blue points for that family are not given in Fig. 3-right. Can you please elaborate on what type of grids, (and most importantly why) the Lloyd could be a limiting factor? and please report the aspect ration location in Fig 3-right.
3.	For fig 1 what is the greedy dominance factor?
4.	In fig 2 can we also see the greedy baseline?


**Time Spent Reviewing:**

3

---

> ### Author Response · Authors · 2021-08-10
> **Adding details for Lloyd algorithm and its O(n) time complexity; addressing raised points related to figures.**
>
> 1- We will add a paragraph to Sec 4.3 to detail the Lloyd aggregation algorithm. Moreover, we will provide the full algorithm and will show why Lloyd achieves O(n) time complexity in the appendix. The Lloyd algorithm is indeed essentially k-means, but on a graph rather than in the usual R^n setting. Operating on a graph (using Bellman-Ford and Floyd-Warshall for distance computations) means the algorithm has the performance bounds we need. We realize that we did not explain this clearly enough in the submitted manuscript and we will clarify these details.
>
> 2- Indeed, the aspect ratio family is (by far) the most challenging family in our tests and, in fact, lies outside of the underlying theory from MacLachlan and Saad [17].  On the right side of Figure 3, results for this family were not reported since the theory doesn’t guarantee convergence of AMG using the interpolation algorithm that we consider here. To make this clearer, we will provide more detail on the underlying theory, including a discussion of the key assumptions and limitations from MacLachlan and Saad [17], in the appendix. We note, however, that our direct interest in this paper is in the question of coarsening these graphs, and not of optimizing convergence of the multigrid solvers (which is complementary work that is underway).  From this view, the aspect ratio family still represents a challenging test for our algorithm, in which it still achieves better measure (f-fraction) than the previous method. It is also noteworthy that the Lloyd algorithm is not the limiting factor in this case.
>
> 3- The dominance factor we have set for our study is theta=0.56, which is mentioned in Sec 3.1, line 141. Nevertheless, we will add this to figure captions for clarity.
>
> 4- We will provide the greedy baseline to figure 2.

---

### Official Review · Reviewer_984W · 2021-07-16

**Rating:** 4
**Confidence:** 3

**Summary:**

The authors try to use reinforcement learning to investigate large sparse linear systems of equations. Meanwhile, they introduce the graph model here.

**Limitations And Societal Impact:**

Theories related to their experiments still need to improve.

**Main Review:**

Using reinforcement learning to solve large sparse linear systems of equations is a very interesting problem. The authors introduce the graph and GNN as the tools to investigate them. However, the quality is not enough and the content is not clear. The experiments look very good, but the theories are not very clear. If the authors can show strong theories, I would like to change my grade. Furthermore, the authors still need to show the significance of their work to the readers.

**Time Spent Reviewing:**

25

---

> ### Author Response · Authors · 2021-08-10
> **Adding all the related theories to the appendix; notes on significance of the work.**
>
> The theoretical underpinnings of the work are covered in the cited reference [17] by MacLachlan and Saad.  We recognize from this comment that we did not make this clear enough in the initial submission, and will add details about this theory in the appendix to the manuscript.  Moreover, we will provide a more thorough introduction to algebraic multigrid (AMG) solvers as well.
>
> Regarding the significance of our work, we will further emphasize the fact that our study is the first learning-based method, with a convergence guarantee, for coarsening of grids within AMG. AMG coarsening with guaranteed convergence is a constraint programming problem which is best modeled using a temporal setting; hence, we have proposed a reinforcement learning method, strictly outperforming existing algorithms. Furthermore, our method achieves linear time complexity for the NP-hard problem of AMG coarsening, and it can be evaluated on any arbitrary planar grid (the localization is generally applicable to any size and shape planar grid), strictly achieving better performance than the previous greedy algorithm. Until proposing this method, a learning-based AMG coarsening method was the missing link for having an end-to-end learning-based AMG solver, given there are learning-based algorithms for determining AMG relaxation schemes and interpolation/prolongation operators.

---

> ### Comment · Area_Chair_WbLM · 2021-08-23
> **To: reviewer 984W**
>
> Dear Reviewer, are you satisfied with the answer?
> Can you elaborate what kind of theory you expect in the original review.

---

### Decision · Program_Chairs · 2021-09-28

**Decision:**

Accept (Poster)

**Comment:**

## Original AC meta-review

This paper proposes a systematic way of computing optimal coarsening for algebraic multigrid (AMG) algorithms by using a) connection of this problem to graph algorithm b) formulation of the optimal selection as a game c) solution of the RL problem using graph neural networks. The resulting algorithm is compared against classical greedy algorithm and beats it on a wide range of meshes. One of the important contributions is the usage of LLoyd aggregation that allows the learned coarsening to be computed in linear complexity. Most of the reviewers agree that the approach is interesting algorithmically and promising in practice. Some concern was raised about theoretical contribution, but it is not so clear to me where this should go: the equivalence to RL task is shown, and how good the RL algorithms solves problem is quite difficult to show.

For the experiments, as it was noted in the reviews, the resulting convergence is only shown for 2D Laplace problem, which can be considered as a minus for the paper. However, the framework that are proposed do not depend on the type of the PDE, so this would have been a nice addition, but not change the main message of the paper.

To summarize, this is one of the first works where RL approach seems to provide a practical tool for constructing AMG precondition and can be important for the field.

## SAC meta-review (the AC's recommendation was borderline)

This paper had quite opposite recommendations from the reviewers so I had to check more carefully how to calibrate them with respect to the ones on other accepted / rejected papers. I also read the paper. While I agree that this paper presents an interesting line of research that should be presented to the machine learning community, the current submission did not appear ready yet for publication at NeurIPS. In particular, as mentioned by some reviewers, the authors should make their paper more self-contained in terms of many non-standard topics in the machine learning community (like Lloyd's aggregation, which is not described at all even in an appendix). The work should also be better related to the current machine learning literature: for example, RL has already been used to solve several combinatorial algorithms (see [A],[B] e.g.), how does their approach to solve (4) fit and compare with this literature? (this was not mentioned at all in the current submission) Finally, comparing only with the greedy algorithm of [17] appears limited for such a niche area, especially since the authors already mentioned that their approach should be more effective than simulated annealing [18] in lines 147-148, but yet do not demonstrate this at all empirically or theoretically. Including such a comparison would improve the significance of the work.

I encourage the authors to take these comments in consideration for a resubmission.

[A] Bengio, Y., Lodi, A., & Prouvost, A. (2021). Machine learning for combinatorial optimization: a methodological tour d’horizon. European Journal of Operational Research, 290(2), 405-421. (on arXiv since 2018)
[B] Mazyavkina, N., Sviridov, S., Ivanov, S., & Burnaev, E. (2021). Reinforcement learning for combinatorial optimization: A survey. Computers & Operations Research, 105400. (on arXiv since 2020)

**Consistency Experiment:**

NeurIPS has a long history of experimentation. In 2014, NeurIPS ran an experiment in which 10% of submissions were reviewed by two independent committees to quantify the randomness in the review process. This year, we repeated a variant of this experiment to see how the quality of the review process has changed over time.  This paper was part of the experiment and was therefore assigned to two committees (consisting of reviewers, an Area Chair, and a Senior Area Chair) that reached independent decisions.  If both committees made the same recommendation, this recommendation was followed. If a single committee recommended acceptance, the paper was accepted (with the exception of a few cases in which the other committee identified what we considered a fatal flaw, e.g., an error in a key result).

This copy’s committee reached the following decision: **Reject**

The other committee assigned to the paper recommended **Accept (Poster)**.  You can find the other set of reviews, along with any follow up discussion with the authors here:
https://openreview.net/forum?id=WcY6S6PDuly